# Hypericin Suppresses SARS-CoV-2 Replication and Synergizes with Antivirals via Dual Targeting of RdRp and 3CLpro

**DOI:** 10.3390/microorganisms13051004

**Published:** 2025-04-27

**Authors:** Helena da Silva Souza, Jéssica Santa Cruz Carvalho Martins, Thiagos das Chagas Sousa, Saiqa Sardar, Natalia Fintelman-Rodrigues, Lina Silva-Trujillo, Thiago Moreno Lopes e Souza, Marilda Mendonça Siqueira, Jorge Hernandes Fernandes, Aline da Rocha Matos

**Affiliations:** 1Laboratory of Respiratory Viruses, Exanthematics, Enteroviruses and Vital Emergencies, Oswaldo Cruz Institute, Fiocruz, Rio de Janeiro 21040-900, Brazil; ssouza.helena@gmail.com (H.d.S.S.); jessicadl_10@hotmail.com (J.S.C.C.M.);; 2Laboratory of Immunopharmacology, Centro de Pesquisa, Inovação e Vigilância em COVID-19 e Emergências Sanitárias, Oswaldo Cruz Institute, Rio de Janeiro 21040-361, Brazil; nataliafintelman@gmail.com (N.F.-R.); marsilva1293@gmail.com (L.S.-T.); souzatml@gmail.com (T.M.L.e.S.); 3Center for Technological Development in Health, National Institute for Science and Technology on Innovation on Neglected Diseases Neglected Populations, Oswaldo Cruz Foundation, Rio de Janeiro 21040-900, Brazil; 4Laboratório de Química e Função de Proteínas e Peptídeos, Centro de Biociências e Biotecnologia, Universidade Estadual do Norte Fluminense (UENF), Campos dos Goytacazes 28013-602, Brazil

**Keywords:** SARS-CoV-2, COVID-19, variants, hypericin, RdRp, 3CLpro, metadynamics

## Abstract

The continuous emergence of SARS-CoV-2 variants underscores the need for novel antiviral candidates. Hypericin (HY), a compound derived from *Hypericum perforatum*, exhibited potent in vitro activity against SARS-CoV-2 in Vero E6 cells, with low cytotoxicity (CC_50_ > 200 nM). HY showed no significant activity against Influenza A (H1N1) or dengue virus serotype 2, supporting its selective action. Antiviral effects were most evident when HY was administered post-infection, in a concentration-dependent manner, while cellular pretreatment or viral pre-incubation produced limited effects. Notably, HY also displayed virucidal activity, significantly reducing viral titers at 4 °C, 22 °C, and 37 °C. Combination treatments with remdesivir or nirmatrelvir enhanced antiviral efficacy by 50–70% relative to monotherapy, depending on compound concentration. Molecular simulations revealed stable interactions with conserved residues in RdRp and 3CLpro, suggesting a low risk of resistance. Together, these findings highlight the potential of HY as a selective antiviral and virucidal agent against SARS-CoV-2, particularly in combination regimens.

## 1. Introduction

Coronavirus disease 2019 (COVID-19) emerged as a global public health crisis caused by the Severe Acute Respiratory Syndrome Coronavirus 2 (SARS-CoV-2), highlighting the vulnerability of the human population to new infectious pathogens [1]. SARS-CoV-2 is an RNA virus from the Coronaviridae family [2]. Its rapid spread resulted in the emergence of viral variants, such as alpha, beta, gamma, delta, and omicron, which were promptly identified through genomic surveillance [3,4,5]. The variants’ genetic changes predominantly occurred in the spike gene, but additional genomic regions, which are targets of available antivirals, were also altered [6,7,8].

Vaccines are essential to reduce the disease burden, utilizing different types of immunogens [9]. However, its unequal distribution and the need for adequate infrastructure compromise vaccination coverage [10]. The efficacy of vaccines may vary among at-risk groups and against viral variants [11,12]. Consequently, the search for effective therapies is essential for treating infected patients and reducing mortality, highlighting the importance of antivirals and immunomodulators in disease management, particularly in vulnerable populations [13,14]. While vaccines are essential, new therapies are crucial for treatment of individuals infected with emerging variants.

COVID-19 clinical presentation varies widely, ranging from asymptomatic cases to development of acute respiratory distress syndrome (ARDS) and death [5,15]. At-risk groups for disease severity include older individuals and those with comorbidities, such as hypertension and diabetes [16]. The cytokine storm that occurs after viral infection is associated with the disease severeness [17,18]. The pathophysiology of COVID-19 is clinically characterized by different stages of infection, each requiring specific therapeutic approaches. In the disease acute phase, during mild to moderate infection, SARS-CoV-2 levels and replication in the upper respiratory tract are high. This early phase, characterized by active viral replication, represents a critical window where antiviral therapies can be most effective in reducing viral load, potentially limiting disease progression. As the infection progresses, in the further immunopathological phase, the exacerbated inflammatory response can lead to severe complications, such as hypoxemia, thrombosis, leading to multiple organ failure that increases the need for ventilatory and circulatory support. In the critical phase, immunomodulators and intensive care are fundamental, with prophylactic anticoagulation often indicated [19,20]. Understanding these mechanisms is crucial for developing therapies to eradicate the virus and mitigate inflammatory damage [21,22].

The pathophysiology of COVID-19 is clinically characterized by different stages of infection, each requiring specific therapeutic approaches. In the disease acute phase, during mild to moderate infection, SARS-CoV-2 levels and replication in the upper respiratory tract are high. This early phase, characterized by active viral replication, represents a critical window where antiviral therapies can be most effective in reducing viral load, potentially limiting disease progression. As the infection progresses, in the further immunopathological phase, the exacerbated inflammatory response can lead to severe complications, such as hypoxemia, thrombosis, leading to multiple organ failure that increases the need for ventilatory and circulatory support. In the critical phase, immunomodulators and intensive care are fundamental, with prophylactic anticoagulation often indicated.

SARS-CoV-2 replication involves critical stages in which the viral non-structural proteins (NSPs), including RNA-dependent RNA polymerase (RdRp) and the main protease (Mpro), also termed as 3-chymotrypsin-like proteases (3CLpro), play essential roles [22]. RdRp is responsible for the replication and transcription of viral RNA, forming a tripartite complex with NSP-7 and NSP-8 and interacting with NSP-14, which provides proofreading exonuclease activity. 3CLpro acts in the cleavage of viral polyproteins 1a and 1b into NSPs, further activating the additional proteins necessary to assemble and release new viral particles [23]. 3CLpro and RdRp are highly conserved among different viruses, especially within the CoV genera and positive-sense RNA viruses. This conservation makes them attractive and promising targets for developing effective antivirals against SARS-CoV-2 and other pathogens [24].

Since the onset of the COVID-19 pandemic, therapeutic strategies against SARS-CoV-2 infection have evolved significantly [13]. Currently, the antiviral landscape for COVID-19 treatment includes remdesivir (RDV), which has received full approval from the FDA, while nirmatrelvir-ritonavir (NMV/r) and molnupiravir have received Emergency Use Authorization [25,26]. Specifically, molnupiravir’s EUA is more restricted, being authorized only when other authorized COVID-19 treatment options are not accessible or clinically appropriate. This regulatory distinction is important as it reflects differences in efficacy and safety profiles. However, while monoclonal antibodies were initially granted emergency use authorization, their approvals were revoked due to the emergence of omicron subvariants, which demonstrated resistance to these therapies [26]. In parallel, host-targeted therapies and immunomodulatory agents aimed at modulating the hyperinflammatory response that is observed in severe cases of the disease are currently under active investigation [27].

Currently, monoclonal antibodies were initially granted emergency use authorization, but their approvals were subsequently revoked due to the emergence of omicron subvariants that demonstrated resistance to these therapies [28].

Despite the availability of some antiviral agents, there are growing concerns about resistance, especially in the context of emerging SARS-CoV-2 variants [28]. Studies have reported the potential for resistance development, including clinical and virological relapses in immunocompromised patients treated with RDV monotherapy [6]. In vitro evidence further suggests that specific mutations in SARS-CoV-2 could confer resistance to both RDV and NMV [29,30]. Emerging evidence indicates that using these antivirals could accelerate viral evolution, potentially giving rise to resistant variants and diminishing their therapeutic efficacy [30]. Consequently, developing a broader array of antiviral agents with enhanced potency, immunomodulatory effects, and anti-inflammatory properties is essential for expanding the therapeutic armamentarium against SARS-CoV-2 infection [31].

Hypericin (HY) is an anthraquinone with photosensitizing properties predominantly found in plants of the species *Hypericum perforatum*. When HY is protected from light, it exhibits therapeutic activities, including against viruses such as hepatitis C, HIV, influenza A, and infectious bronchitis virus (IBV), an avian CoV [32,33,34,35,36]. Additionally, HY has been described as having antitumor properties [37,38]. A previous report from our group showed that HY inhibited SARS-CoV-2 replication in vitro in a concentration-dependent manner with no significant cytotoxic effects [39]. Further studies corroborated these findings, reporting the potent action of HY against SARS-CoV-2 in vitro, suggesting its antiviral and virucidal activities [40,41]. Therefore, we aimed to evaluate and confirm HY’s antiviral effects in cells infected with SARS-CoV-2 variants, assessing its efficacy before and after infection. Additionally, we investigated its activity both as a monotherapy and in combination with other antivirals. To further explore its mechanism of action, we performed molecular simulations to analyze its interactions with viral RdRp and 3CLpro.

## 2. Materials and Methods

### 2.1. Cell Culture

We used Vero E6 and Vero CCL-81 cell lines, derived from the African green monkey kidney, which are permissive for SARS-CoV-2 infection and replication, as previously demonstrated by our group and others [39,42,43]. The basic culture medium employed was Dulbecco’s Modified Eagle Medium (DMEM), containing D-glucose (4.5 g/L) and L-Glutamine (3.9 mM), supplemented with 10% Fetal Bovine Serum (FBS) and a Penicillin-Streptomycin 100× solution (with final concentrations of 100 U/mL and 100 μg/mL, respectively). Both cell lines were stored in a CO_2_ incubator, with conditions of 37 °C, 5% CO_2_, and 95% relative humidity. Additionally, molecular tests based on PCR were conducted periodically to detect potential mycoplasma contamination in the cell cultures, which were all negative during our experiments.

### 2.2. SARS-CoV-2 Isolates

This study used distinct and relevant SARS-CoV-2 variants, including lineage B.1, gamma, delta, and omicron (BA.1 and BA.5), according to the official WHO nomenclature. Viral isolation was performed from human respiratory secretion samples received in the Respiratory Viruses, Exanthematous Viruses, Enteroviruses, and Viral Emergencies Laboratory (LVRE)/Fiocruz, which is a WHO reference lab, as part of the Brazilian surveillance system. All procedures related to patient samples were approved by the research ethics committee of the Oswaldo Cruz Institute (registration CAAE 68118417.6.0000.5248).

Viral isolates were propagated to produce a working stock. The B.1, gamma, delta, and omicron BA.1 and BA.5 isolates were subjected to genomic sequencing, and their sequences were deposited in the GISAID platform with the following accession numbers: EPI_ISL_414045, EPI_ISL_1219135, EPI_ISL_2645414, EPI_ISL_9225756, and EPI_ISL_12838778, respectively.

All procedures related to viral isolation and subsequent manipulations were conducted in the multi-user facilities of a biosafety level 3 (BSL3) lab at Fiocruz in compliance with WHO guidelines.

### 2.3. Compounds

The HY compound used in this study is derived from *Hypericum perforatum* and was purchased from Sigma-Aldrich (Merck KGaA, Darmstadt, Germany). The compound, supplied in a lyophilized form, was resuspended with 100% Dimethyl Sulfoxide (DMSO) (Merck KGaA, Darmstadt, Germany), resulting in a working stock that was stored according to the manufacturer’s instructions. As a photosensitive compound, HY was permanently protected from light. Stock solutions were stored in amber tubes and wrapped in aluminum foil. During all procedures, including dilution in culture medium and treatment of cells, HY was handled inside a biosafety cabinet with the internal light turned off. The light remained off throughout the entire experimental protocol, including infection and treatment steps, to ensure full protection from light exposure. RDV and NMV were commercially acquired (MedchemExpress, Monmouth Junction, NJ, USA), supplied in lyophilized form, and further resuspended in 100% DMSO (Merck KGaA, Darmstadt, Germany), resulting in stocks that were stored according to the manufacturer’s instructions.

### 2.4. Viral Titration

The viral titers obtained in the supernatant of the treated cells were determined using the lysis plaque assay method. Briefly, Vero CCL-81 cells were seeded for 24 h to achieve monolayers with 100% confluence. After that, cells were infected with a serial dilution of each aliquot and overlayed with agarose (Invitrogen, Thermo Fisher Scientific, Waltham, MA, USA) solution. Moreover, 48 h post-infection (hpi) cells received a second overlay, composed of a neutral red (Sigma-Aldrich, St. Louis, MO, USA) solution. Lysis plaques were visually counted after 24 h using a white light transilluminator (Avantor, Radnor, PA, USA).

### 2.5. HY IC_50_ Determination In Vitro

Vero E6 cells were seeded 24 h before the experiment. HY was used in final concentrations of 0.002–200 nM. For the infection assay, culture media was removed from the cells’ monolayers, followed by washing them with PBS (Gibco, Waltham, MA, USA). A multiplicity of infection (MOI) of 0.01 was used for infection, which was performed for 1 h in the incubator. The plates were gently agitated every 10 min to facilitate viral adsorption. After that, the inocula were removed, and HY was added. Cell supernatants were collected 48 hpi and stored at −80 °C for subsequent viral titration. A concentration-response curve was generated, and the concentration required to inhibit viral replication by 50% (IC_50_) was determined through nonlinear regression using GraphPad Prism (Version 8).

### 2.6. Cytotoxicity Assays

To evaluate the cytotoxicity of HY, Vero E6 and MDCK cells were seeded in 96-well plates (2 × 10^4^ cells/well) and treated with increasing concentrations of the compound for 48 h at 37 °C in a 5% CO_2_ humidified atmosphere. Cell viability was assessed using the MTT assay, and results were expressed as the percentage of viable cells compared to the DMSO-treated control. For Vero E6 cells, the concentration range tested was from 2 nM to 200 μM, with 10-fold serial dilutions (log_10_ interval), aiming to cover a broad range of sensitivity due to the high tolerance previously reported for this cell line. In contrast, for MDCK cells, a narrower range of 12.5 nM to 200 nM was used, with 2-fold serial dilutions (log_2_ interval), based on preliminary data indicating increased sensitivity to the compound in this lineage. The 50% cytotoxic concentration (CC_50_) was determined by nonlinear regression analysis using GraphPad Prism software.

### 2.7. Evaluation of HY Antiviral Activity Against SARS-CoV-2 Variants

To evaluate the impact of post-infection treatment with HY on the replication of SARS-CoV-2 variants, including B.1, gamma, delta, and omicron (BA.1 and BA.5), we conducted an assay in infected cells, as previously described. HY was employed at concentrations of 20 nM and 200 nM for post-infection treatment over 48 h. All procedures related to cell infection and the final steps of viral suspension collection, titration, and statistical analysis were conducted using the same protocols previously mentioned.

### 2.8. Determination of HY Pre- and Post-Infection Antiviral Effect

Vero E6 cells were seeded one day before the experiment. Treatment was performed in the following conditions: total treatment; cells pre-infection treatment for 1 h; virus incubation for 1 h at 4, 20, and 37 °C; treatment for 1 h during infection; and treatment post-infection for 48 h. HY was used at a concentration of 200 nM. All procedures related to cell infection and the final steps of viral supernatant collection, titration, and statistical analysis were conducted as previously described.

### 2.9. Drug Combination Assays

To compare the in vitro efficacy of HY against SARS-CoV-2 with RDV and NMV, we performed post-infection treatment assays. To investigate the impact of double/triple drug combination therapy, HY was used in a range of concentrations (1–100 nM), chosen as having suboptimal antiviral effect. The RDV and NMV concentrations used (2000 and 4000 nM, respectively) also demonstrated previous suboptimal anti-SARS-CoV-2 effects. All procedures related to cell infection and the final steps of viral suspension collection, titration, and statistical analysis were conducted as previously described for infections with the variants.

### 2.10. Evaluation of HY Antiviral Effect Against Influenza A(H1N1)pdm09 and DENV-2

Vero E6 cells and Madin-Darby canine kidney cells (MDCK) were cultured in DMEM high glucose supplemented with 100 μg/mL streptomycin, at 37 °C in a humidified atmosphere containing 5% CO_2_. For viral replication inhibition assays, MDCK and Vero E6 cells (2 × 10^4^ cells/well) were seeded in 96-well plates and infected after 24 h with influenza A(H1N1)pdm09 virus (MOI 0.01, for 1 h) or DENV-2 (MOI 0.1, for 1 h) at 37 °C. Following viral adsorption, the inoculum was removed, and the cells were treated with HY (2 or 20 nM) for 24 h (influenza) or 48 h (DENV-2). As a positive control, influenza viruses were also treated with oseltamivir at 10 µM. After incubation, cell culture supernatants were collected for viral RNA quantification by real-time RT-PCR. Total RNA was extracted using the Maxwell^®^ RSC Instrument (Promega, Madison, WI, USA), according to the manufacturer’s instructions. Quantitative RT-PCR was performed using a GoTaq^®^ Probe 1-Step RT-qPCR system (Promega, Madison, WI, USA) in a QIAquant 96 detection system (QiAgen, Hilden, Germany). Amplifications were carried out in 10 µL reaction mixtures containing 7.5 µL reaction mix buffer, 0.3 µL enzyme mix, 20 µM each primer, 10 µM probe, and 5 µL of RNA template. Primers, probes, and cycling conditions are described elsewhere [44,45]. The standard curve method was employed for viral quantification. Viral inhibition was expressed as the percentage of viral load reduction relative to untreated virus-infected controls.

### 2.11. SARS-CoV-2 3CLpro and RdRp Initial Structures

Our experiments used the most promising results obtained by our previous docking experiments for SARS-CoV-2 3CLpro and RdRp in a complex with HY [39]. The complexes are formed by the protein structure of SARS-CoV-2 3CLpro (PDB: 6LU7) [46] and the SARS-CoV-2 RdRp domain (7bv2) [47]. The structure of these proteins was obtained from the PDB database, and AutoModel [48] was used to complete the regions without structural information through homology modeling. In this process, the four initial residues were deleted for better quality structural information and packing in our model. In both structures, all water and hetero-atoms were removed for docking experiments. In all the previous docking and actual simulation experiments, Mg^2+^ ions were maintained in the active site of the SARS-CoV-2 RdRp.

### 2.12. Molecular Simulation of the 3CLpro and RdRp in Complex with HY

Starting from the protein-ligand complex with the best interaction energy obtained in previous docking experiments, molecular simulations were performed to define the stability of the ligand positioning in the complex and the definition of the protein residues essential for maintaining the ligand binding.

The RdRp-HY complex is composed of 1120 amino acids (911 from chain A, 114 from chain B, 67 from chain C, and 28 from chain D). A HY ligand (Hyp), two Ca^2+^ ions, and three Mg^2+^ ions (two of them essential cofactors for the polymerization reaction catalyzed by RdRp) were initially placed in a triclinic simulation box (4.05 × 43 × 28.1 nm) with periodic conditions. Approximately 38,260 explicit water molecules and 53 Na^+^ and Cl^−^ ions were added to simulate the physiological conditions of 150 mM salt. The final simulation system contained 132,812 atoms.

Following the same steps, the final simulation 3CLpro-HY system contained 35.064 atoms. The initial complex, composed of 306 residues and a Hyp, was accommodated in a triclinic simulation box (53.25 × 67.86 × 87.37) with 10.090 water molecules and 28 Na^+^ and Cl^−^ ions (150 mM salt). The obtained systems were parameterized in Desmond-GPU-Maestro 2022-4 (www.deshawresearch.com, Assessed on 30 June 2023) using the OPLS-2005 force field and SPCE water model [49]. The simulation system was equilibrated to 5 ns in a mixed NVT/NPT system [50]. Three to five data production simulations replicates were performed for 50 to 200 ns on a Linux server using an NPT system at 300 K and 1 bar of pressure using a variant of the Nosé-Hoover thermostat [51] and the Martyna-Tobias-Klein algorithm as barostat [52]. The simulation time step used was 2.0 fs in the RESPA integration method, and the electrostatic forces were calculated using the u-series method [53] with a cutoff of 0.9 nm. Each replicate run goes through several stages according to the Maestro instructions as detailed in the Appendix A.

### 2.13. Metadynamics Experiments

Metadynamics experiments were performed using Desmond-GPU-Maestro (www.deshawresearch.com, assessed on 30 June 2023) to estimate the interaction energy between HY and SARS-CoV-2 3CLpro and RdRp enzymes. The systems described above, after equilibration for 10 ns in conditions of explicit water and 150 mM salts, were subjected to metadynamics using the dedicated panel in Maestro (https://support.schrodinger.com/s/article/1621, assessed on 30 June 2023). The complex, already solvated, as described above, was loaded into the Metadynamics panel to prepare the input files for the experiments. The energy variations of the complex were collected due to the variation of two collective variables (CV), the distances between the centers of mass of the protein residues and the ligand.

For the RdRp-HY complex, the CV distance was defined between the center of mass of residues Arg552 (CV1) and Lys846 (CV2) to the center of mass of HY, with values of 2.6 and 3.6 nm, respectively, as the maximum collection wall. For the 3CLpro-HY complex, the CV of distance was defined from the center of the mass of HY to the center of the CYs38 (CV1) and Thr26 (CV2) with the collection wall values of 2.0 and 1.6, respectively. The Gaussian width and height were defined as 0.05 Å and 0.01 kcal/mol, respectively. The Gaussian injection time interval was set at 0.09 ps. The simulation temperature and pressure were defined as described above for other simulations. However, in the data production, the Martyna-Tobias-Klein dynamics [51] of the NPT ensemble with metadynamics is performed at 310 K without restrictions for 50 to 100 ns. Three replicates were collected. 

The variables of each experiment were collected and plotted in a two-dimensional graph in Maestro to define wells of the complex’s best free energy (ΔG). The structures with the lowest ΔG value were analyzed, and the observed protein-ligand contacts were compared to those obtained in previous docking experiments and molecular simulations. Other parameters in the metadynamics experiments are detailed in the Appendix A.

### 2.14. Bioinformatic Analysis of 3CLpro Conservation

To evaluate the conservation of the amino acid region 187–192 of the SARS-CoV-2 3CLpro, a key target in our study, we performed a variant screening using the CoV-Spectrum platform [54]. This tool accesses SARS-CoV-2 genomic data from the Global Initiative on Sharing All Influenza Data (GISAID) database and enables filtering by mutation, time period, and geographic origin. The analysis was carried out in April 2025, including all sequences deposited globally since 2020.

### 2.15. Statistical Analysis

Statistical analyses were conducted using GraphPad Prism version 8.0.1 (GraphPad Software Inc., San Diego, CA, USA), employing one-way ANOVA with Dunnett’s or Sidak’s multiple comparisons tests for the concentration-response assays of HY against the variants, as well as for the pre- and post-infection treatment experiment. For the combined treatment with RDV and/or NMV, a two-way ANOVA with Sidak’s multiple comparisons was used. Non-linear regression was employed to determine HY IC50. Experimental results were considered significant when * *p* < 0.05, ** *p* < 0.01, *** *p* < 0.001, and **** *p* < 0.0001.

## 3. Results

### 3.1. HY Inhibits SARS-CoV-2 in a Concentration-Dependent Manner

Previous data from our group already demonstrated a concentration-dependent reduction in SARS-CoV-2 RNA load in the supernatant of HY-treated cells [36]. However, these earlier results involved SARS-CoV-2 quantification through viral RNA measurement in the cell supernatant. To confirm these effects with a more precise methodology, we herein determined the HY IC_50_ by viral titration in a plaque assay. Therefore, we observed decreased SARS-CoV-2 (B.1 lineage isolate) titers correlated with increasing HY concentrations. The analysis indicated that HY at 0.2 nM and 2 nM significantly reduced viral replication, resulting in mean viral titers of 2.6 × 10^6^ and 1.7 × 10^6^ PFU/mL, respectively, compared to the control that showed mean titers of 4 × 10^6^ PFU/mL (*p* = 0.0089 and *p* < 0.0001, respectively). Higher concentrations (20 nM and 200 nM) achieved complete inhibition of viral replication, eliminating viral titers (*p* < 0.0001) (Figure 1A). As a control, RDV (5000 nM) was also found to completely inhibit viral replication (*p* < 0.0001) (Figure 1A). Therefore, our analysis showed that HY IC_50_ was approximately 0.35 nM (Figure 1B).

In addition, HY demonstrated low cytotoxicity in both Vero E6 and MDCK cell lines. The calculated CC_50_ value for HY in Vero E6 cells was 316.8 ± 86.5 nM, whereas in MDCK cells, no cytotoxic effects were observed up to 200 nM (CC_50_ > 200 nM). These results suggest that the antiviral activity of HY occurs at concentrations that are not cytotoxic to host cells, indicating a favorable selectivity index.

### 3.2. HY Shows No Antiviral Activity Against RNA Viruses Such as Influenza A and DENV-2

To evaluate whether HY exerts a broader antiviral effect, we tested its activity against two other RNA viruses: influenza A(H1N1)pdm09 subtype and DENV-2. The two concentrations of HY (2 and 20 nM) were selected based on their significant antiviral activity against SARS-CoV-2 to evaluate potential broad-spectrum antiviral properties. Different timepoints for viral quantification (24 h for Influenza A versus 48 h for DENV-2) were chosen to capture the optimal window for viral replication in each virus model. In viral replication inhibition assays, treatment with HY (2 and 20 nM) did not significantly reduce viral RNA levels in comparison to untreated controls (Figure 2). While influenza-infected cells showed an average reduction of approximately 50% in viral RNA at HY concentrations tested, the difference was not statistically significant. No inhibitory effect was observed for DENV-2. These results suggest that the antiviral effect of HY may be specific to SARS-CoV-2.

### 3.3. HY Displays Broad-Spectrum Antiviral Activity Against SARS-CoV-2 Variants

To evaluate the antiviral activity of HY against SARS-CoV-2 relevant variants, we tested the efficacy of the compound against older and more recent VOCs, such as gamma, delta, and omicron variants (BA.1 and BA.5). HY concentration (2 and 20 nM) was selected based on previous dose-response experiments, corresponding approximately to the IC_50_ and near-IC_90_ values, respectively, in a non-cytotoxic range. Our data indicated that the highest concentration of HY tested (20 nM) completely inhibited the replication of variants B.1, gamma, delta, omicron BA.1, and omicron BA.5 (*p* < 0.0001, *p* = 0.0001, *p* < 0.0001, *p* = 0.0014, and *p* < 0.0001, respectively) (Figure 3). At the lower tested concentration (2 nM), a significant reduction in viral titers was also observed for all tested variants, including B.1, gamma, delta, omicron BA.1, and omicron BA.5, when compared to the untreated control (*p* = 0.0002, *p* = 0.0002, *p* = 0.0006, *p* = 0.0014, and *p* < 0.0001, respectively). Specifically, the mean viral titers in the untreated control were [4.6 × 10^6^ PFU/mL for B.1], [4.4 × 10^6^ PFU/mL for gamma], [4.5 × 10^6^ PFU/mL for delta], [4.5 × 10^6^ PFU/mL for BA.1], and [7.2 × 10^6^ PFU/mL for BA.5], whereas treatment with 2 nM HY reduced these values to [5 × 10^5^ PFU/mL], [3.6 × 10^5^ PFU/mL], [1.6 × 10^6^ PFU/mL], [1.3 × 10^6^ PFU/mL], and [3.7 × 10^6^ PFU/mL], respectively. These reductions correspond to a decrease of 88.38%, 91.16%, 63.69%, 73.48%, and 47.87% in viral replication, respectively (Figure 3). These findings demonstrate that HY exhibits potent antiviral activity against multiple SARS-CoV-2 variants, achieving complete inhibition at 20 nM and significantly reducing viral titers even at a lower concentration (2 nM), highlighting its potential as a broad-spectrum antiviral candidate.

### 3.4. HY Exhibits Post-Infection and Virucidal Activity Against SARS-CoV-2

We performed a time-of-addition assay to investigate the fundamental mechanism of action of HY against SARS-CoV-2 in vitro. HY concentration (200 nM) was selected based on previous dose-response experiments to ensure maximum antiviral effect for clear mechanistic determination. Our results showed that both total treatment (i.e., treatment before, during, and after infection) and post-infection treatment alone completely inhibited viral replication (*p* < 0.0001) (Figure 4). In contrast, treatment applied only before infection had no antiviral effect, as viral titers remained comparable to those of the untreated control group, with mean values of 3.8 × 10^6^ PFU/mL vs. 4.1 × 10^6^ PFU/mL, respectively.

To provide a comprehensive comparison across all treatment conditions, we conducted statistical analyses among the different groups (Figure 4). Total treatment resulted in significantly lower viral titers compared to pre-treatment of cells (mean titer: undetectable vs. 3.7 × 10^6^ PFU/mL; *p* < 0.0001) and virus pre-incubation at 4 °C, 22 °C, and 37 °C (mean titers: 2.2 × 10^6^, 2.5 × 10^6^, and 2.4 × 10^6^ PFU/mL, respectively; *p* = 0.0012, *p* = 0.0003, and *p* = 0.0005). No significant difference was observed between total and post-infection treatment (both with undetectable titers, *p* > 0.9999), nor between total and during-infection treatment (mean titer: undetectable vs. 1.3 × 10^6^ PFU/mL; *p* = 0.0801).

Pre-infection treatment was also significantly less effective than both post-infection (*p* < 0.0001) and during-infection treatment (*p* = 0.0004), confirming that HY exerts its antiviral activity predominantly after viral entry.

Interestingly, pre-incubation of the virus with HY at different temperatures (4 °C, 22 °C, and 37 °C) led to significant reductions in viral titers compared to the control (*p* = 0.0010, 0.0043, and 0.0026, respectively), with mean values ranging from 2.2 × 10^6^ to 2.5 × 10^6^ PFU/mL. However, no significant differences were observed among the temperature groups themselves, suggesting that HY’s virucidal activity is not temperature-dependent.

Additionally, treatment during infection resulted in a substantial decrease in viral replication (mean titer: 1.3 × 10^6^ PFU/mL; *p* < 0.0001), though not as complete as the inhibition seen with post-infection or total treatment. Our findings suggest that HY exerts its strongest antiviral effect when administered post-infection, completely inhibiting viral replication. While HY also exhibits virucidal activity, its efficacy is not influenced by temperature variations. These results indicate that HY primarily acts after viral entry, highlighting its potential as a therapeutic agent against SARS-CoV-2.

### 3.5. Combination Therapy of HY with RDV and NMV Enhances Viral Inhibition

To assess the potential of HY in combination with the currently approved antiviral drugs RDV and NMV, we compared its antiviral activity in monotherapy and combined treatments. The combination of HY (1 nM) with RDV (2 µM) resulted in a significant reduction in viral titers compared to HY monotherapy, with a 70.59% greater reduction than single-drug treatments (mean titer: 8.3 × 10^5^ PFU/mL vs. 2.8 × 10^6^ PFU/mL; *p* < 0.001). Similarly, the combination of HY at 10 nM with RDV led to a 53.33% additional reduction in viral titers compared to HY monotherapy (mean titer: 7.0 × 10^5^ PFU/mL vs. 1.5 × 10^6^ PFU/mL; *p* < 0.01). However, at 100 nM, HY alone already achieved the maximum antiviral effect, showing no further benefit when combined with RDV at this concentration (mean titer: 3.3 × 10^4^ PFU/mL vs. undetectable) (Figure 5A). It is important to note that treatment with RDV alone (2 µM) showed titers of 2.8 × 10^6^ PFU/mL.

Additionally, the combination of HY (1 nM and 10 nM) with NMV (4 µM) also resulted in a statistically significant reduction in viral titers, in comparison with HY monotherapy (mean titer: 9.7 × 10^5^ PFU/mL vs. 2.8 × 10^6^ PFU/mL and 5.0 × 10^5^ PFU/mL vs. 1.5 × 10^6^ PFU/mL, respectively; *p* < 0.0001 and *p* < 0.01, respectively), with an incremental reduction of 65.88% and 66.66% compared to HY monotherapy. As observed with RDV, the combination of HY at 100 nM with NMV did not enhance the antiviral effect beyond what was already achieved by HY alone at this concentration (mean titer: 6.6 × 10^4^ PFU/mL vs. 3.3 × 10^4^ PFU/mL) (Figure 5B). It is important to mention that treatment with NMV alone (4 µM) showed titers of 1.4 × 10^6^ PFU/mL.

Furthermore, the triple combination of HY (1 nM and 10 nM) with RDV (2000 nM) and NMV (4000 nM) demonstrated an additional reduction in viral titers by 65.88% and 62.22%, respectively, compared to HY monotherapy (mean titer: 9.7 × 10^5^ PFU/mL vs. 2.8 × 10^6^ PFU/mL and 5.7 × 10^5^ PFU/mL vs. 1.5 × 10^6^ PFU/mL, respectively; *p* < 0.0001 and *p* < 0.01, respectively) (Figure 5C). These findings indicate that HY, when used in combination with RDV and NMV, enhances antiviral activity compared to monotherapy, particularly at lower concentrations (1–10 nM). However, at 100 nM, HY alone is sufficient to achieve maximum viral suppression (mean titer: 1.0 × 10^5^ PFU/mL vs. 3.3 × 10^4^ PFU/mL), suggesting a concentration threshold beyond which combination therapy does not provide additional benefits.

These results support the potential of HY as an effective component in combination treatment strategies against SARS-CoV-2.

### 3.6. Molecular Simulations and Metadynamics of SARS-CoV-2 3CLpro in Complex with HY

The ligand demonstrated remarkable stability and minimal displacement during the simulations, with consistently low root mean square deviation (RMSD) values across all simulation replicates (Appendix A). This suggests that the ligand’s conformation, as determined through initial docking studies [39], represents an energetically favorable binding mode (Figure 6A,B). HY remains bound to the active site of 3CLpro, partially occupying the substrate recognition region and positioning itself as a competitive inhibitor based on its stable structural orientation within the complex (Figure 6B).

HY maintains a stable interaction profile with 3CLpro through hydrogen bonds mediated by crucial residues, including Asp187, Arg188, Thr190, and Gln192. Hydrophobic contacts with Met49 and Met165 further stabilize these interactions on one side of the binding pocket and water-mediated bridges with Glu166 on the opposite side (Appendix A).

These residues consistently interact with the ligand throughout most of the simulation across all replicates (Appendix A), and no significant conformational rearrangements were observed in the analyzed trajectories. Additional contacts were observed sporadically or in only one replicate and were considered less critical or of minimal contribution to ligand recognition and complex stability. Only interactions maintained for over 50% of the simulation trajectory were classified as essential for the recognition and binding process (Appendix A). Throughout its interaction with the enzyme, HY remains partially inserted within the active site, avoiding the aqueous phase as much as possible while maintaining partial exposure to the solvent (Figure 6E).

The metadynamics simulations were conducted using the same system as the molecular dynamics experiments, with the equilibrium time reduced to 10 ns. Unlike the linear progression of traditional molecular dynamics simulations, metadynamics employs an adaptive technique that forces the system to explore regions outside its energy minimum. During these simulations, two collective variables (CVs) were monitored: the distances between the HY, Cys38, and Thr26 centers of mass. These variables were used to analyze the binding affinity within the complex. The results across all replicates consistently revealed a single basin of the lowest free energy (ΔG), with an estimated value ranging from −15 to −17 kcal/mol (Figure 6D). This energy landscape reflects the stable accommodation of HY within the enzyme’s active site, engaging the identical residues identified in the molecular dynamics simulations (Figure 6C and Appendix A). HY forms several hydrogen bonds with crucial residues, including Thr190, Asn142, and Asp187, while its hydrophobic interactions are stabilized by Met49 and Met165 (Figure 6E).

The convergence of binding site analysis from both metadynamics and molecular simulations, highlighting the identical group of critical residues involved in ligand interactions, strongly supports the hypothesis that HY fits precisely into the 3CLpro active site, acting as a competitive inhibitor. This behavior is likely a contributing factor to its observed virucidal activity.

To assess whether the conservation of the region encompassing amino acids 187–192 of the SARS-CoV-2 3CLpro protease, which would interact with HY, across circulating viral lineages, we analyzed sequence data available on the CoV-Spectrum platform (https://cov-spectrum.org, assessed on 9 April 2025), which integrates global SARS-CoV-2 genomes primarily deposited in the GISAID database. As of April 2025, only 51 viral sequences worldwide have been reported to carry mutations within this region: 3 at position 188, 11 at 189, 1 at 190, 10 at 191, and 26 at 192, with no mutations detected at position 187. These mutations are rare and sporadically distributed across different geographic regions and time points, suggesting that this region remains highly conserved.

### 3.7. Molecular Simulations and Metadynamics of SARS-CoV-2 RdRp in Complex with HY

Minor conformational adjustments of HY at the binding site were observed in all simulations of the RdRp-HY complex, previously characterized in docking experiments [39]. Despite this, HY exhibited minimal displacement, reflected in low RMSD values across simulations (Appendix A). Positioned near the catalytic Mg^2+^ ions within the RdRp active site and partially occupying the RNA binding region, HY is stabilized above Trp797 through π-π stacking interactions between their aromatic rings (Figure 7B,C). This orientation is further reinforced by hydrogen bonds formed with residues in the Asp801-Glu808 loop, along with electrostatic interactions involving Glu799. On the opposite side, additional stabilizing hydrogen bonds were observed with Ser604 and Asp605 (Figure 7C,E and Appendix A). In all replicas, only interactions sustained for over half of the simulation trajectories were considered critical for the RdRp-HY interaction (Appendix A).

During its interaction with the enzyme, HY remained embedded within the hydrophobic cleft adjacent to the active site (Figure 7B,C), avoiding aqueous exposure as much as possible. Concurrently, metadynamics experiments were performed to explore HY’s binding affinity further. Two collective variables (CVs) were monitored: the distances between the centers of mass of HY, Lys157, and Arg552 as the ligand moved within the x-y plane of the active site. The resulting free energy profile revealed a single basin with an estimated energy of −12 to −13 kcal/mol (Figure 7D), representing the hydrophobic stabilization of HY within the enzyme’s cleft near the active site. Similar to the findings in the HY-3CLpro simulations, molecular simulations and metadynamics experiments consistently identified the identical key residues involved in complex formations (Figure 7C and Appendix A).

## 4. Discussion

Since the emergence of the COVID-19 pandemic, intensive efforts have been made to develop effective treatment options against this disease, especially in the most severe cases. One of the main goals was to identify suitable compounds that would be capable of inhibiting virus replication and further disease complications and, therefore, reduce the burden on healthcare systems [55]. In this context, our group previously screened compounds approved by international agencies as candidate therapeutic agents for repurposing against SARS-CoV-2, through the SBVS experiment, specifically focusing on compounds that would bind 3CLpro and RdRp. As a result, HY was one of the candidates identified, and we showed that it could act as an anti-SARS-CoV-2 replication inhibitor in vitro [39].

Herein, we initially confirmed our previous results by demonstrating a concentration-dependent reduction in the replication of SARS-CoV-2 in vitro. However, we employed distinct methodologies for measuring the in vitro antiviral effect in the previous and current reports by our group. Our previous findings were methodologically based on the quantification of viral RNA in the cell supernatant of treated cells after infection by using quantitative real-time PCR [39]. In the current report, we measure the HY antiviral effect by applying the lysis plaque method for viral titration. The reason for the alternation of methods was the evolution and dissemination of SARS-CoV-2 titration methodologies throughout the pandemic. In the meantime, between our two reports, additional literature data indicated that, under similar in vitro infection conditions and HY treatment, the compound blocked SARS-CoV-2 propagation [40]. It is important to point out that in both previous reports, no cytotoxic effect was observed for HY-treated cells in the range of concentrations that we tested.

To explore whether the antiviral effect of HY extends beyond SARS-CoV-2, we evaluated its activity against two other RNA viruses: influenza A(H1N1)pdm09 and DENV-2. Interestingly, HY did not inhibit DENV-2 replication, and only a modest, non-significant reduction of influenza A virus replication was observed. These findings suggest that HY does not exert a general antiviral effect across RNA viruses. Given our molecular docking and metadynamics analyses suggesting that HY binds with high affinity to the SARS-CoV-2 RdRp, one possible explanation for this selectivity is a preferential interaction with structural features unique to the CoV polymerase complex. Despite functional conservation, the RdRp of SARS-CoV-2 differs structurally and mechanistically from the polymerases of negative-sense viruses such as influenza A (which uses a heterotrimeric complex) and from the flaviviral RdRp in DENV-2. These differences may account for the lack of inhibition observed in these models. Further comparative structural and biochemical studies could help clarify whether HY selectively targets conserved domains or residues specific to the SARS-CoV-2 RdRp.

To further evaluate host cell safety, cytotoxicity assays were conducted in the same cell lines used for infection. The CC_50_ of HY in Vero E6 cells was 316.8 ± 86.5 nM, and no significant cytotoxicity was observed in MDCK cells at concentrations up to 200 nM. Considering that the antiviral IC_50_ values observed for SARS-CoV-2 and influenza virus were lower than these thresholds, HY exhibits a favorable therapeutic window.

We further explored HY’s activity against other relevant and more recent SARS-CoV-2 variants, such as gamma, delta, and omicron BA.1 and BA.5 sub-lineages. Both concentrations that we tested showed a very high impact in reducing viral loads for all variants, which had also been shown for alpha, beta, and delta variants [40]. Together, these results are complementary, highlighting HY’s broad-spectrum antiviral activity against SARS-CoV-2 variants. This suggests that the compound’s mechanism of action, which involves binding to conserved regions of the virus rather than mutation-prone residues, is unlikely to be affected by ongoing viral evolution.

To investigate the fundamental mechanism of HY antiviral effect in vitro, we performed three distinct time-of-addition assays: (i) pre-treatment of cells, (ii) co-treatment during infection, and (iii) post-infection treatment. (i) In the pre-treatment condition, cells were treated with HY prior to viral inoculation to investigate whether the compound could block viral receptors on the host cell surface or induce antiviral responses such as interferon production. However, no significant reduction in viral titers was observed when compared to the untreated control, indicating that HY does not exert a protective or stimulatory effect on host cells prior to infection. (ii) In the co-treatment assay, HY was added simultaneously with the virus. This condition revealed a pronounced antiviral effect, comparable to the virucidal assay, suggesting that HY may bind directly to viral particles, impairing their infectivity. This supports the hypothesis that HY interferes with viral entry, possibly by blocking fusion events between the viral and cellular membranes. This is consistent with previous reports describing non-specific interactions between HY and viral envelopes or envelope proteins, where the compound showed activity against enveloped viruses but not against non-enveloped ones such as adenovirus or poliovirus [56] (iii) In the post-infection treatment, HY was added after viral entry into host cells. This condition also led to a strong inhibition of viral replication, corroborating previous findings that showed a significant reduction in viral nucleocapsid protein expression when HY was administered post-infection [39,40]. These results suggest that HY not only acts at early stages of infection but may also interfere with intracellular processes during viral replication. It has been proposed that HY may alter protein-protein interactions or the subcellular localization of host proteins and enzymes [57,58].

In addition, we performed a virucidal assay in which viral particles were incubated with HY prior to cell infection. A strong virucidal activity was observed, and this effect was not influenced by the tested temperature range, indicating that HY remains stable and active across different thermal conditions. While some antiviral agents are sensitive to temperature variations, affecting their stability and interaction with viral components [59,60], HY maintained its bioactivity [61]. Furthermore, in silico analyses have shown that HY binds to the viral membrane with high affinity and interacts with key functional regions of the SARS-CoV-2 S protein. Its insertion into the viral envelope, especially within the S protein internal cavity in the pre-fusion conformation, may hinder the structural rearrangements necessary for membrane fusion and viral entry [41,62].

The specificity of antiviral compounds is a critical factor in assessing their translational potential. In our study, we observed that pre-treatment of host cells with HY for 1 h prior to viral infection did not reduce viral replication, further supporting the hypothesis that HY does not interfere directly with cellular components. To assess potential off-target interactions, we considered the structural similarity between viral targets and host proteins. Notably, SARS-CoV-2 RdRp and 3CLpro lack close homologs in the human proteome. The viral RdRp is structurally and functionally distinct from human RNA polymerases, and 3CLpro, a cysteine protease with a chymotrypsin-like fold, does not share significant homology with human proteases. Previous structural and docking studies have shown minimal interaction between inhibitors of these viral enzymes and human proteins [63], supporting the selectivity of such antiviral agents. Together with our experimental results, these findings suggest that HY is unlikely to exert its antiviral effects via host modulation and presents a low risk of off-target cytotoxicity.

These findings indicate that HY, when used in combination with RDV or NMV, enhances antiviral activity compared to monotherapy, particularly at lower concentrations (1–10 nM). However, at 100 nM, HY alone is sufficient to achieve maximum viral suppression, suggesting a concentration threshold beyond which combination therapy does not provide additional benefits. These results support the potential of HY as an effective component in combination treatment strategies against SARS-CoV-2.

Furthermore, our findings demonstrate that HY (1–10 nM), when used in combination with RDV or NMV, increased viral reduction compared to monotherapy. It is important to reinforce that HY, RDV, and NMV may have distinct antiviral mechanisms of action. RDV mainly targets viral RdRp by inhibiting this protein and being incorporated into the viral RNA [64]. In contrast, NMV, which is a covalent peptidomimetic inhibitor, binds to the active site of the viral protease 3CLpro through a nitrile moiety, forming a reversible covalent bond with its catalytic residue Cys145 [25]. This distinction in their mechanisms suggests that HY may act through a complementary pathway, potentially targeting a different stage or target in the viral life cycle. The observed effect in combination treatments supports this hypothesis, indicating that HY might enhance viral inhibition by interfering with processes not directly affected by RdRp or 3CLpro inhibition, thereby amplifying the overall antiviral response.

Previously reported in silico analyses demonstrated that HY has a high affinity for viral proteins 3CLpro and RdRp. In our current analysis, we propose that HY blocks RdRp through a hydrophobic interaction with the Trp797 residue and inhibits 3CLpro by blocking its active site, acting as a competitive inhibitor. It has also been reported that HY interacts with the palm domain of RdRp near its catalytic center, blocking the natural ribonucleotide pathway at the enzyme’s active site—an essential step for viral replication [39,65,66]. Similar findings have been obtained using virtual screening methodologies, either targeting the S protein [58], evaluating anti-HIV bioactive compounds against SARS-CoV-2 RdRp [65,66], or identifying HY as an inhibitor of other viral proteins, such as NSP-14 [67]. Additionally, multiple studies have reported a strong interaction between HY and 3CLpro [65,68,69]. One study, using Förster Resonance Energy Transfer (FRET) experiments, classified HY as a weak inhibitor of SARS-CoV-2 3CLpro [70]. Furthermore, HY has also demonstrated antiviral activity against alphaCoVs by inhibiting 3CLpro [71].

It is essential to correlate our findings with potential mutations in the RdRp and 3CLpro enzymes in the analyzed variants. For 3CLpro, the P132H mutation, present exclusively in the BA.1 and BA.5 variants, has been characterized as a unique alteration of omicron. Located 22 Å from the catalytic Cys145, this mutation is situated between the catalytic and dimerization domains, not causing direct structural changes in the active site [72]. Through molecular dynamics simulations, we observed that HY forms a small network of hydrogen bonds with highly conserved residues (187–192) inside the *Coronaviridae* (taxid:11118) (Appendix A), primarily with Gln192 and Asp187. Thus, it is likely that the antiviral action of HY is not affected by the P132H mutation in the omicron variants, as the substance would act in a distinct region. In the case of RdRp, the most critical hydrophobic contact is the π-stacking with Trp979, also conserved within the *Coronavidae* (Appendix A). The region 187–192 surface pocket of SARS-CoV-2 3CLpro is located adjacent to the catalytic site and may influence protease function or substrate accessibility. Importantly, 3CLpro is known to be a highly conserved enzyme among CoVs, due to its essential role in viral replication. Previous analyses of this protein were conducted as a comprehensive genetic surveillance and demonstrated strong conservation at both the sequence and structural levels [73]. In line with this, our targeted mutational survey using the CoV-Spectrum tool revealed that mutations in the 187–192 region are rare among globally circulating SARS-CoV-2 genomes. This supports the hypothesis that the antiviral effect of hypericin is unlikely to be broadly compromised by naturally occurring mutations in this region. Consequently, based on the in silico data, it is possible that HY may have multimodal antiviral functions and may act as a pan-CoV antiviral agent, given that 3CLpro and RdRp are highly conserved across different CoV families. Further studies are needed to characterize the mechanism of action of HY thoroughly, and this will be the next step in our experimental interest.

The surveillance of mutations associated with antiviral resistance and the development of new combination therapies are crucial for optimizing treatment and ensuring the efficacy of approved COVID-19 antiviral drugs, such as RDV, Paxlovid, and Molnupiravir, which have active registration in Brazil [74]. Although these drugs are effective against SARS-CoV-2 infection, recent studies reveal the potential emergence of resistance, exploring various pathways in vitro. The identified mutations provide a solid foundation for deepening the understanding of resistance mechanisms, as well as assisting in the development of new antivirals and identifying factors that promote or prevent this resistance [29,30,75,76].

Despite the robust antiviral activity demonstrated by HY in vitro and its potential synergistic effects in combination with approved antivirals, this study has some limitations. All experiments were performed using Vero cells, which, although widely used for SARS-CoV-2 studies due to their high permissiveness, do not fully recapitulate the complexity of human respiratory epithelium or immune responses. Future studies using human-derived cell lines or organoid models are necessary to better characterize the effects of HY in a more physiologically relevant context. In addition, although our molecular simulations provided insights into the potential interaction of HY with viral proteins, these findings were not yet experimentally validated by direct binding assays or structural studies. These aspects will be addressed in future work to strengthen the mechanistic understanding and translational relevance of HY as an antiviral agent.

## 5. Conclusions

HY exhibits significant antiviral activity against SARS-CoV-2, demonstrating both virucidal properties and inhibition of viral replication across multiple variants, including gamma, delta, and omicron sub-lineages. Our findings suggest that HY maintains its efficacy despite the diversity of mutations accumulated over viral evolution, reinforcing its potential as a broad-spectrum antiviral agent. Additionally, HY enhances antiviral activity when used in combination with RDV or NMV, particularly at lower concentrations, suggesting a complementary mechanism of action that could be explored in future therapeutic strategies.

Our in vitro and in silico analyses indicate that HY interacts with key viral proteins, such as RdRp and 3CLpro, through specific molecular interactions that likely contribute to its antiviral effect. Furthermore, its virucidal activity appears to be linked to interactions with the viral envelope, potentially preventing viral fusion and entry into host cells. Importantly, HY was well tolerated in cell models, with no significant cytotoxicity observed in the tested concentration range.

Given the ongoing emergence of SARS-CoV-2 variants and the potential for antiviral resistance, further studies are needed to elucidate the full mechanism of action of HY, its efficacy against other coronaviruses, and its role in combination therapies. These findings contribute to the growing body of evidence supporting HY as a promising candidate for future antiviral development, with potential applications beyond SARS-CoV-2.

## Figures and Tables

**Figure 1 microorganisms-13-01004-f001:**
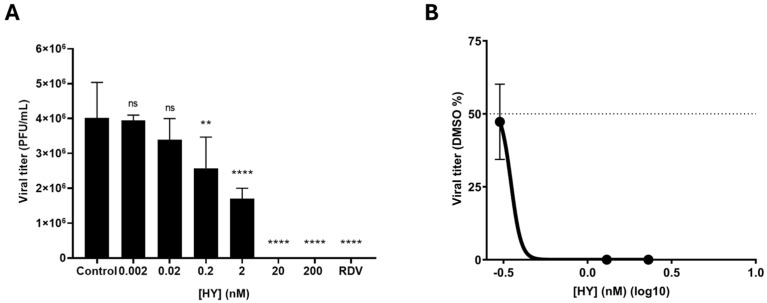
HY reduces SARS-CoV-2 replication in vitro in a concentration-dependent manner. (**A**) Vero E6 cells were infected with SARS-CoV-2 and further treated with HY (0.002–200 nM) for 48 h. Viral titers were quantified in cell supernatants collected at 48 hpi and subjected to plaque assay in Vero-CCL81 cells. Results are expressed as PFU/mL (mean ± SD). The positive control (RDV) was used at 5000 nM. Data are presented as absolute viral titers to demonstrate the magnitude of viral reduction across different concentrations. Statistical analysis was performed using one-way ANOVA with Dunnett’s multiple comparisons test, comparing each value to the DMSO control. ** *p* < 0.01, **** *p* < 0.0001, n = 4 (number of independent replicates for each experiment). hpi—hours post-infection, SD—standard deviation, RDV—Remdesivir, ns—not significant. (**B**) Representation of the concentration-response curve with viral titers expressed as % of the DMSO control (Mean ± SD). Each experimental data point was normalized to the average titer of the DMSO control and plotted against HY concentration using a logarithmic scale on the *x*-axis to better visualize the full range of tested concentrations. Nonlinear regression was performed using the obtained data to determine the HY IC_50_. The dotted line indicates 50% of the control value, which defines the IC_50_.

**Figure 2 microorganisms-13-01004-f002:**
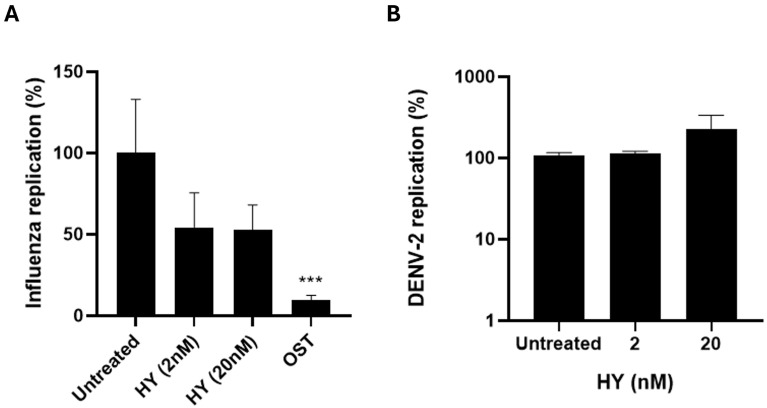
HY does not inhibit replication of Influenza A or DENV-2 in vitro. (**A**) MDCK cells (for influenza A(H1N1)pdm09) and (**B**) Vero E6 cells (for DENV-2) were infected with MOI 0.01 or MOI 0.1, respectively, and treated with HY (2 and 20 nM) for 24 h (Influenza A) or 48 h (DENV-2). As a positive control, influenza viruses were also treated with oseltamivir at 10 µM. Viral RNA levels in cell culture supernatants were quantified by real-time RT-PCR. Results are expressed as % of viral RNA relative to the untreated control (mean ± SD). The data are presented as a percentage of the control. Statistical analysis was performed using one-way ANOVA with Dunnett’s multiple comparisons test, comparing each treatment to the untreated control. *** *p* < 0.001 n = 3 (independent experiments).

**Figure 3 microorganisms-13-01004-f003:**
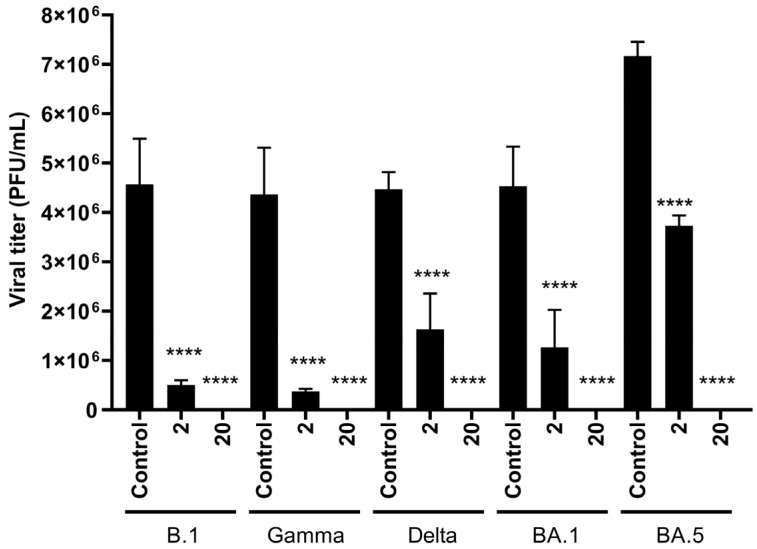
HY exhibits antiviral activity against SARS-CoV-2 variants in a concentration-dependent fashion. Vero E6 cells infected with SARS-CoV-2 variants B.1, gamma, delta, omicron BA.1, and omicron BA.5 were treated with HY (2 nM and 20 nM). Viral titers were quantified in cell supernatants collected at 48 hpi and subjected to plaque assay in Vero-CCL81 cells. Results are expressed as PFU/mL (mean ± SD). The data are presented as absolute viral titers (PFU/mL). Statistical analysis was performed using one-way ANOVA with Sidak’s multiple comparisons test, comparing each value to the DMSO control. **** *p* < 0.0001, n = 3 (number of independent replicates for each experiment). hpi—hours post-infection, SD—standard deviation.

**Figure 4 microorganisms-13-01004-f004:**
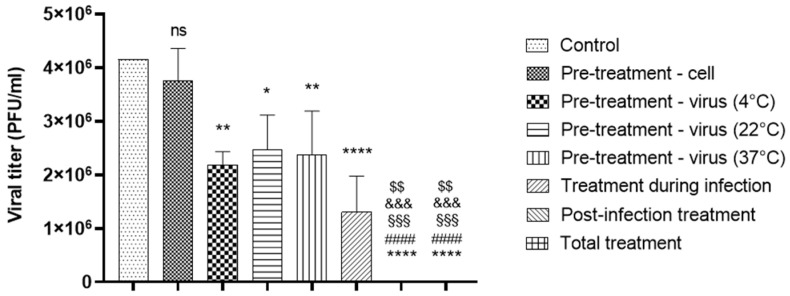
Time of addition evaluation of HY against SARS-CoV-2. SARS-CoV-2-infected Vero E6 cells were treated with different treatment schemes using HY at 200 nM. Viral titers were quantified in cell supernatants collected at 48 hpi and subjected to plaque assay in Vero-CCL81 cells. Results are expressed as PFU/mL (mean ± SD). The data are presented as absolute viral titers (PFU/mL). Each bar represents a specific treatment condition. Statistical analysis was performed using one-way ANOVA with Dunnett’s multiple comparisons test. * *p* < 0.05, ** *p* < 0.01, **** *p* < 0.0001 versus DMSO; #### *p* < 0.0001 versus pre-treatment—cell; $$ *p* < 0.01 versus virus 4 °C; &&& *p* < 0.001 versus virus 20 °C; §§§ *p* < 0.001 versus virus 37 °C, n = 3 (number of independent replicates for each experiment). hpi—hours post-infection, SD—standard deviation, ns—not significant.

**Figure 5 microorganisms-13-01004-f005:**
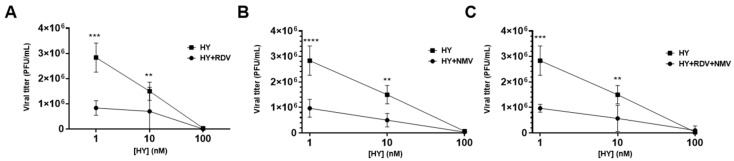
Combination of HY with RDV and NMV. Antiviral activity of HY against SARS-CoV-2 in vitro, in combination with RDV (**A**), NMV (**B**), and RVD + NMV (**C**). SARS-CoV-2-infected Vero E6 cells were submitted to different treatment schemes by using HY at 1, 10, and 100 nM and/or RDV (2 µM) and NMV (4 µM). Viral titers were quantified in cell supernatants collected at 48 hpi and subjected to plaque assay in Vero-CCL81 cells. Results are expressed as PFU/mL (mean ± SD). The data are presented as absolute viral titers (PFU/mL). Statistical analysis was performed using two-way ANOVA with Sidak’s multiple comparisons test, comparing each value of the double or triple combination titer with the respective concentration of the HY monotherapy. ** *p* < 0.01, *** *p* < 0.001, and **** *p* < 0.0001, n = 3 (number of independent replicates for each experiment). hpi—hours post-infection, SD—standard deviation.

**Figure 6 microorganisms-13-01004-f006:**
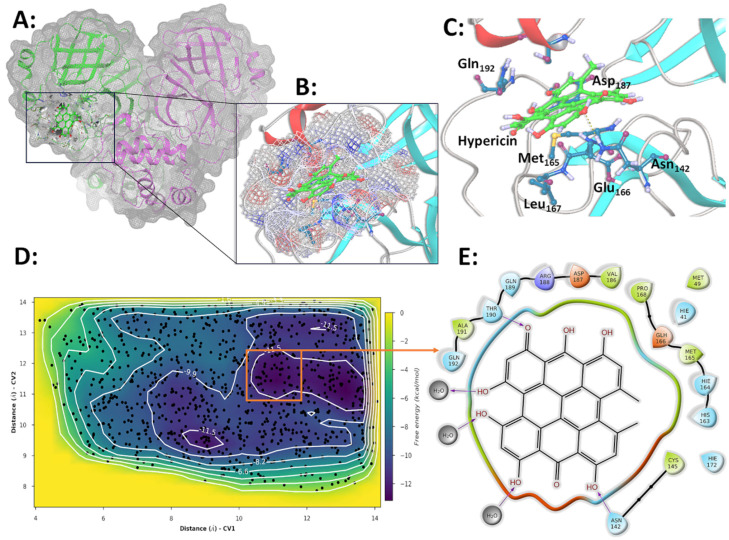
Representation of important SARS-CoV-2 3CLpro contacts with HY, obtained in simulation and metadynamics experiments. (**A**) General view of the 3CLpro-HY complex. The protein contact surface is represented in blue for positive and red for negative electrostatic. HYinhibitor is represented in van der Waals surface and colored green for C and red for O atoms. (**B**) HY is inserted in the enzyme’s active site. (**C**) Most important residues contacting HY in simulation experiments. (**D**) Metadynamics results indicate a global energy minimum of −17 kcal/mol in the 3CLpro-HY complex. A bi-dimensional representation of the variables collected by CV1 and CV2 was obtained using Maestro. (**E**) Most important interactions in the global minima complex obtained in metadynamics experiments, converging to the same essential contact residues obtained in simulation experiments. Polar contributions are represented in light blue, electrostatic interactions are in red and blue, and hydrophobic in green. (**A**–**C**) Complex representations were obtained using Pymol version 2.3.0.

**Figure 7 microorganisms-13-01004-f007:**
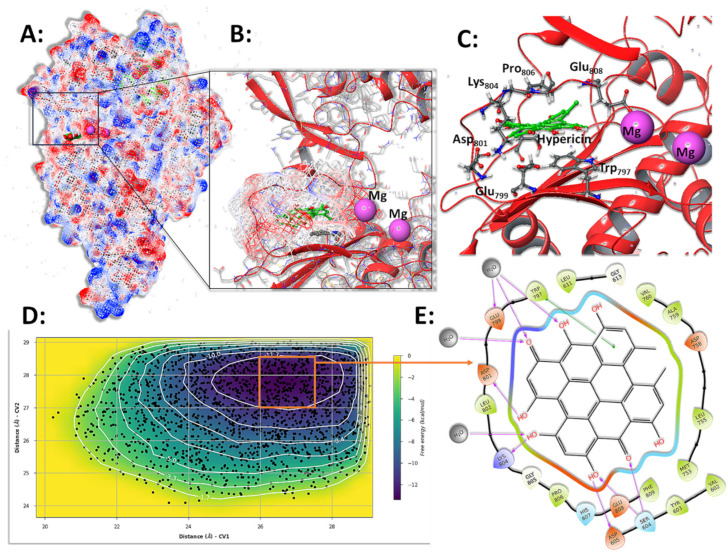
Representation of important SARS-CoV-2 RdRp contacts with HY, obtained in simulation and metadynamics experiments. (**A**) General view of the RdRp-HY complex. (**B**) HY is inserted near the enzyme’s active site in a hydrophobic pocket. (**C**) Important residues contacting HY in simulation experiments. (**D**) Metadynamics results indicate a global energy minimum of −12.5 kcal/mol in the RdRp-HY complex. (**E**) Important interactions in the global minima complex obtained in metadynamics experiments converge to the same contact residues. Complex representations were obtained using Pymol version 2.3.0.

## Data Availability

The original contributions presented in this study are included in the article/Appendix A. Further inquiries can be directed to the corresponding author.

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
