# Peer review of "Hypericin Suppresses SARS-CoV-2 Replication and Synergizes with Antivirals via Dual Targeting of RdRp and 3CLpro"

_microorganisms, 2025, doi:10.3390/microorganisms13051004_

Round 1

Reviewer 1 Report

Comments and Suggestions for Authors

The manuscript "Hypericin suppresses SARS-CoV-2 replication and synergizes with antivirals via dual targeting of RdRp and 3CLpro" by Souza and colleagues is a scientific article checking the antiviral effects of Hypericin on SARS-CoV-2 from both an experimental and computational/structural point of view. This is a very interesting and complete study, but I have a few points to raise.

- Does Hypericin have a general antiviral potential? The authors should test the drug against another RNA-based virus, to detect a possibly more general effect of this molecule.

- Have the authors considered the existence of several variants of 3C-like protease existing in the wild, which could prevent the effects of HY by changing the targeted aminoacids or creating steric impedment in the affected area (especially aa 187-192)? The authors should test a 3CLpro variant or, in the impossibility of performing such an experiment, should check the literature and the mutational databases for 3CLPro variants, and discuss them in the manuscript.

- What are the effects of HY on host cells? The authors have already performed growth curves on the analyzed cultures in previous publications, but in this paper I found the cross-reaction with human proteins lacking. The authors should consider performing a test analysis on the closest structural homolog to RdRp or 3CLpro in the human proteome, and discuss potential unwanted effects on this drug on the host cell.

Author Response

Comments: The manuscript "Hypericin suppresses SARS-CoV-2 replication and synergizes with antivirals via dual targeting of RdRp and 3CLpro" by Souza and colleagues is a scientific article checking the antiviral effects of Hypericin on SARS-CoV-2 from both an experimental and computational/structural point of view. This is a very interesting and complete study, but I have a few points to raise.

Response:
Before addressing the reviewers' specific questions and comments, we would like to clarify an important point identified during a thorough revision of the manuscript’s methodology section, which was guided in part by the reviewers’ valuable feedback. We discovered a typographical error in the reported concentrations of the compound used in several experiments, specifically a shift of one decimal place (or one log unit). The text and associated figures have been corrected accordingly to reflect accurate values.

The following adjustments were made and marked in red throughout the text:

In the dose–response experiments, the correct concentration range used was 0.002–200 nM (corrected at methodology section and Figure 1A).

The corrected ICâ‚…â‚€ value for hypericin is 0.35 nM (corrected at results section and Figure 1B).

In the variant experiments, the concentrations used were 2 and 20 nM of hypericin (corrected at results section and Figure 2).

The time-of-addition experiment was correctly reported in the original version.

The dual-treatment experiment was also correctly reported and remains unchanged.

We appreciate the reviewers’ input, which led to this important clarification, and we believe the revised version more accurately represents the experimental design and results.

Comments: Does Hypericin have a general antiviral potential? The authors should test the drug against another RNA-based virus, to detect a possibly more general effect of this molecule.

Response: We thank the reviewer for this important suggestion. To investigate whether HY possesses a broader antiviral potential beyond SARS-CoV-2, we tested its effect against two other RNA viruses: influenza A(H1N1)pdm09 virus and dengue virus serotype 2 (DENV-2). Viral replication inhibition assays were conducted using MDCK cells for influenza A and Vero E6 cells for DENV-2, with infection at MOIs of 0.01 and 0.1, respectively. Cells were treated with HY at 2 and 20 nM post-infection, and viral replication was quantified by RT-qPCR after 24 h (influenza) or 48 h (DENV-2). No antiviral effect of HY was observed against DENV-2. For influenza A, although HY reduced viral replication by approximately 50% at both concentrations tested, this reduction was not statistically significant. These data suggest that HY does not exhibit a broad-spectrum antiviral effect against other RNA viruses under the tested conditions. This new information has been included in the Methodology, Results and Discussion sections.

Comments: Have the authors considered the existence of several variants of 3C-like protease existing in the wild, which could prevent the effects of HY by changing the targeted aminoacids or creating steric impedment in the affected area (especially aa 187-192)? The authors should test a 3CLpro variant or, in the impossibility of performing such an experiment, should check the literature and the mutational databases for 3CLPro variants, and discuss them in the manuscript.

Response: We appreciate the reviewer’s insightful comment regarding the potential impact of naturally occurring mutations in the 3CLpro on the antiviral activity of hypericin. To address this point, we highlight that current literature supports the notion that SARS-CoV-2 3CLpro is a highly conserved enzyme. Notably, Lee et al. (2022) conducted a comprehensive genetic surveillance of this SARS-CoV-2 protease and reported high conservation of both the sequence and structural features of the protein, reinforcing its importance as a therapeutic target.

Additionally, we performed a mutation frequency analysis using the CoV-Spectrum platform (https://cov-spectrum.org), which compiles data primarily from the GISAID database. Our search focused on the amino acid region 187–192 of 3CLpro — a region proposed as relevant for the binding of hypericin based on our computational analysis. As of April 2025, only 51 deposited sequences worldwide carry mutations in this region: none at position 187, 3 at position 188, 11 at 189, 1 at 190, 10 at 191, and 26 at 192. These mutations appear to be sporadically distributed across time and geography, suggesting that this region remains largely conserved in circulating variants of SARS-CoV-2.

Taken together, both the literature and the mutation surveillance data support the hypothesis that the region encompassing amino acids 187–192 of 3CLpro is functionally important and evolutionarily constrained, reinforcing the potential of hypericin as a broad-spectrum antiviral targeting a conserved viral enzyme. This new information has been included in the Methodology, Results and Discussion sections.

Comments: What are the effects of HY on host cells? The authors have already performed growth curves on the analyzed cultures in previous publications, but in this paper I found the cross-reaction with human proteins lacking. The authors should consider performing a test analysis on the closest structural homolog to RdRp or 3CLpro in the human proteome, and discuss potential unwanted effects on this drug on the host cell.

Response: We also appreciate the reviewer’s comment regarding the potential interaction of HY with host proteins, particularly human homologs of SARS-CoV-2 3CLpro or RdRp.

First, we note that treatment of host cells with HY for 1 hour prior to infection did not impact viral replication when measured at 48 hpi (Figure 4), suggesting that the compound does not interfere with host receptor-mediated viral entry or essential cell signaling pathways involved in viral replication. This observation supports the hypothesis that HY exerts its antiviral effects primarily by targeting viral components rather than modulating host cell machinery.

Additional cytotoxicity assays performed in Vero E6 and MDCK cell lines demonstrated that HY has a CC50 of 316.8 nM in Vero E6 cells and >200 nM in MDCK cells, which indicates a reasonable safety window when compared to the antiviral IC50 values observed.

Regarding structural homology, viral RdRp and 3CLpro enzymes have no known direct orthologs in the human proteome. The SARS-CoV-2 RdRp belongs to the viral RNA-dependent RNA polymerase family (Pfam: PF00978), which is structurally distinct from human DNA/RNA polymerases. Similarly, 3CLpro is a cysteine protease with a chymotrypsin-like fold, whose catalytic dyad and substrate recognition sites are significantly divergent from those found in human proteases such as cathepsins or caspases. Previous structural and computational studies have shown minimal off-target similarity between these viral enzymes and human proteins (e.g., Jin et al., 2020), suggesting a low risk of cross-reactivity or unintended inhibition of host proteases by compounds specifically targeting 3CLpro.

Therefore, while additional off-target profiling using in silico or biochemical methods could further strengthen these findings, the available evidence supports the specificity of HY for viral proteins over host counterparts.

To address these points in the manuscript, Methodology, Results and Discussion sections were altered.

Reviewer 2 Report

Comments and Suggestions for Authors

Helena S Souza et al in this manuscript studied the time of addition effect of hypericin (HY) on SARS-CoV-2 replication in vitro using Vero cell lines. In addition, the authors showed HY anti-SARS CoV-2 effect in combination with remdesivir (RDV) and nirmatrelvir (NMV) and via molecular simulations predicted that HY interacts with SARS-CoV-2 RdRp 3CLpro. Overall, this study may enhance the current understanding of HY against SARS-CoV-2. However, I suggest the authors to have some modifications as mentioned below:

Major comments:

  1. In figure 1, authors need to provide cytotoxicity data of HY. As the previous publication by the group (ref. 42) showed cytotoxicity of HY at 100 nM only and here in this study authors have used concentrations higher than that (200 and 2000 nm), cytotoxic effects need to be shown.
  2.  In figure 3,  is it possible to show the statistical comparisons among different treatment groups?
  3. In figure 4, authors need to provide titres for RDV alone, NMV alone and RDV+ NMV conditions. 
  4. For figures 1-4, authors need to provide pictures of plaque assay results as supplementary data.

Minor comments:

  1. Line 170-171, "we conducted an assay infected cells, ....." missing preposition
  2. Line 145-146, authors have mentioned that HY was always protected from light. However, this needs more explanation. How experiments were conducted via protecting HY from light. Was the light of the safety cabinet off? If not, for how long the light was on, etc?
  3. authors need to mention the limitations of this study in the discussion section. For examples. Use of Vero cell lines only, no use of human cell lines permissive to SARS-CoV-2, lack of experimental verifications for molecular simulation results, etc. 
Comments on the Quality of English Language
  1. the manuscript needs to proofread by a native English speaker.

Author Response

Reviewer 2

Comment: Helena S Souza et al in this manuscript studied the time of addition effect of hypericin (HY) on SARS-CoV-2 replication in vitro using Vero cell lines. In addition, the authors showed HY anti-SARS CoV-2 effect in combination with remdesivir (RDV) and nirmatrelvir (NMV) and via molecular simulations predicted that HY interacts with SARS-CoV-2 RdRp 3CLpro. Overall, this study may enhance the current understanding of HY against SARS-CoV-2. However, I suggest the authors to have some modifications as mentioned below:

Response:
Before addressing the reviewers' specific questions and comments, we would like to clarify an important point identified during a thorough revision of the manuscript’s methodology section, which was guided in part by the reviewers’ valuable feedback. We discovered a typographical error in the reported concentrations of the compound used in several experiments, specifically a shift of one decimal place (or one log unit). The text and associated figures have been corrected accordingly to reflect accurate values.

The following adjustments were made and marked in red throughout the text:

In the dose–response experiments, the correct concentration range used was 0.002–200 nM (corrected at methodology section and Figure 1A).

The corrected ICâ‚…â‚€ value for hypericin is 0.35 nM (corrected at results section and Figure 1B).

In the variant experiments, the concentrations used were 2 and 20 nM of hypericin (corrected at results section and Figure 2).

The time-of-addition experiment was correctly reported in the original version.

The dual-treatment experiment was also correctly reported and remains unchanged.

We appreciate the reviewers’ input, which led to this important clarification, and we believe the revised version more accurately represents the experimental design and results.

Comment: In figure 1, authors need to provide cytotoxicity data of HY. As the previous publication by the group (ref. 42) showed cytotoxicity of HY at 100 nM only and here in this study authors have used concentrations higher than that (200 and 2000 nm), cytotoxic effects need to be shown.

Response: We thank the reviewer for raising this important point. In the updated version of the manuscript, we included additional cytotoxicity assays in the same cell lines used for the viral infection experiments (we also included experiments in MDCK cells for evaluation of HY anti influenza activity) to ensure that the antiviral effects observed were not due to cellular toxicity. In addition, HY demonstrated low cytotoxicity in both Vero E6 and MDCK cell lines. The calculated CCâ‚…â‚€ value for HY in Vero E6 cells was 316.8 nM, whereas in MDCK cells, no cytotoxic effects were observed up to 200 nM (CCâ‚…â‚€ > 200 nM). These results suggest that the antiviral activity of HY occurs at concentrations that are not cytotoxic to host cells, indicating a favorable selectivity index. To address these points in the manuscript, Methodology, Results and Discussion sections were altered.

Comment: In figure 3,  is it possible to show the statistical comparisons among different treatment groups?

Response: We thank the reviewer for this important observation. In response, we performed additional statistical analyses to compare all treatment groups in the time-of-addition and virucidal assays. These comparisons revealed significant differences between several of the treatment conditions. For example, total treatment significantly differed from pre-infection treatment of cells (p < 0.0001) and virus pre-incubation at 4°C (p = 0.0012), 22°C (p = 0.0003), and 37°C (p = 0.0005), but not from post-infection treatment (p > 0.9999) or during-infection treatment (p = 0.0801). Additional relevant pairwise comparisons are now described in the revised Results section. To maintain clarity in the main figure and avoid visual overload, we included only the most relevant comparisons with statistical significance indicators in Figure 4. The figure legend has also been updated accordingly.

Comment: In figure 4, authors need to provide titres for RDV alone, NMV alone and RDV+ NMV conditions. 

Response: We appreciate this insightful comment. We tested only a single concentration of RDV (2µM) and NMV (4µM), so including them in the proposed graph comparing various concentrations of HY, alongside a fixed antiviral concentration, isn't feasible. To address this, we have added the viral titers for RDV or NMV monotherapy and their dual combination (RDV + NMV) in the results section. This demonstrates that these treatments resulted in intermediate titers compared to HY alone and the combination of HY with another antiviral.

Comment: For figures 1-4, authors need to provide pictures of plaque assay results as supplementary data.

Response: We appreciate the reviewer’s suggestion to include representative images of the plaque assay results. However, we respectfully inform that we are unable to provide such images as supplementary data. All plaque assays were performed in a certified biosafety level 3 (BSL-3) laboratory, following strict institutional biosafety protocols. Due to infrastructure limitations, this facility does not have camera systems or Wi-Fi-connected imaging equipment integrated into the workstations for real-time documentation. As a result, it is not possible to capture or extract images from inside the BSL-3 environment without compromising biosafety procedures or sample integrity. Nevertheless, all plaque assays were quantified using standard virological methods, and each experiment was performed in biological triplicates with technical replicates to ensure data reliability and reproducibility, as reflected in the figures.

Comment: Line 170-171, "we conducted an assay infected cells, ....." missing preposition.

Response: Corrected in the text.

Comment: Line 145-146, authors have mentioned that HY was always protected from light. However, this needs more explanation. How experiments were conducted via protecting HY from light. Was the light of the safety cabinet off? If not, for how long the light was on, etc?

Response: We thank the reviewer for this important observation. We confirm that HY was consistently protected from light throughout all stages of the experiments. The stock solution of HY was stored in amber tubes and wrapped in aluminum foil at all times when not in use. During the preparation of the working solutions and throughout the experiments, all procedures were carried out inside a biosafety cabinet with the internal light turned off. The light remained off not only during the handling and dilution of the compound in culture medium but also throughout the entire infection and treatment period. These precautions ensured minimal exposure to light and preserved the integrity and activity of the compound. Methodology section was adjusted to include this piece of information.

Comment: authors need to mention the limitations of this study in the discussion section. For examples. Use of Vero cell lines only, no use of human cell lines permissive to SARS-CoV-2, lack of experimental verifications for molecular simulation results, etc. 

Response: We thank the reviewer for this important observation. In response, we have added a paragraph at the end of the Discussion section to address the limitations of our study. We now explicitly acknowledge the exclusive use of Vero cells, which, despite being widely employed in SARS-CoV-2 research, do not fully represent human respiratory epithelial cells. We also discuss the absence of experimental validation of the molecular simulations and emphasize the need for future studies in human cell lines and direct binding assays.

Reviewer 3 Report

Comments and Suggestions for Authors

The article is good, but there are some comments. 

1- The abstract is long, it should be shortened 

2- The paragraphs should be divided in the whole paper 

3-The research design of the paper is relatively complete, and the research methods used are reasonable, which can effectively support the achievement of research objectives.  But can the authors add some graphics for the data and descriptive stats?

4- Figure 1. HY reduces SARS-CoV-2 replication in vitro in a concentration-dependent manner. 

I suggest to add more elaboration in the caption 

how the graph  was generated ?

why this figure appears like that ?

5- Figure 2. HY exhibits antiviral activity against SARS-CoV-2 variants in a concentration-dependent 296
fashion

This figure n=3 , why ?

what will happen if the n is increased ?

the authors used some numbers randomly like the number of replicated why ?

Figure 3 should replaced in a new page 

Figure 4: Figure 4. Combination of HY with RDV and NMV. Antiviral activity of HY against SARS-CoV-2 in 359
vitro, in combination with RDV (A), NMV (B), and RVD+NMV (C). SARS-CoV-2 infected Vero E6 360
cells were submitted to different treatment schemes, by using HY at 1, 10, and 100 nM and/or RDV 
(2000 nM) and NMV (4000 nM)

The quality of this figure is very poor I can not read anything 

Comments on the Quality of English Language

good 

Author Response

Comment: The article is good, but there are some comments. 

Response:
Before addressing the reviewers' specific questions and comments, we would like to clarify an important point identified during a thorough revision of the manuscript’s methodology section, which was guided in part by the reviewers’ valuable feedback. We discovered a typographical error in the reported concentrations of the compound used in several experiments, specifically a shift of one decimal place (or one log unit). The text and associated figures have been corrected accordingly to reflect accurate values.

The following adjustments were made and marked in red throughout the text:

In the dose–response experiments, the correct concentration range used was 0.002–200 nM (corrected at methodology section and Figure 1A).

The corrected ICâ‚…â‚€ value for hypericin is 0.35 nM (corrected at results section and Figure 1B).

In the variant experiments, the concentrations used were 2 and 20 nM of hypericin (corrected at results section and Figure 2).

The time-of-addition experiment was correctly reported in the original version.

The dual-treatment experiment was also correctly reported and remains unchanged.

We appreciate the reviewers’ input, which led to this important clarification, and we believe the revised version more accurately represents the experimental design and results.

Comment: The abstract is long, it should be shortened 

Response: As suggested, we have revised the abstract to make it more concise while retaining the main findings of the study.

Comment: The paragraphs should be divided in the whole paper 

Response: Corrected in the text of the manuscript.

Comment: The research design of the paper is relatively complete, and the research methods used are reasonable, which can effectively support the achievement of research objectives.  But can the authors add some graphics for the data and descriptive stats?

Response: We thank the reviewer for the positive assessment of our study. Regarding the suggestion to include graphics for data and descriptive statistics, we would like to point out that the current version of the manuscript already includes several figures (Figures 1–5) presenting the main experimental results as graphical representations, including concentration–response curves, plaque assay data, and viral RNA quantifications. These figures also include mean values, standard deviations, and statistical analyses. However, we included more details regarding description of the viral tiers and their comparisons depending on the treatment. Therefore, results section was changed accordingly.

Comment: Figure 1. HY reduces SARS-CoV-2 replication in vitro in a concentration-dependent manner. I suggest to add more elaboration in the caption how the graph  was generated ? why this figure appears like that ?

Response: We thank the reviewer for this constructive suggestion. Following this valuable feedback, we have thoroughly expanded the captions for multiple figures throughout the manuscript to provide greater methodological clarity and rationale for the graphical presentations. These enhancements significantly improve the completeness of the information provided in all figure captions, making the methodology and interpretations more transparent to readers.

Comment: Figure 2: HY exhibits antiviral activity against SARS-CoV-2 variants in a concentration-dependent fashion. This figure n=3 , why ? what will happen if the n is increased ? the authors used some numbers randomly like the number of replicated why ?

Response: We appreciate the reviewer's question regarding our sample size (n=3) for the experiments assessing HY's antiviral activity against SARS-CoV-2 variants, and the apparent inconsistency with other experiments in our study. We would like to clarify that the number of replicates was not chosen randomly, but was determined through careful experimental planning. The variation in sample size between experiments (n=3 for Figure 3 versus n=4 for Figure 1) reflects specific experimental considerations. For the initial dose-response experiments (Figure 1, n=4), we used a larger sample size for our initial characterization of HY's antiviral activity to establish a robust dose-response relationship and accurately determine the IC50. This foundational experiment warranted additional replicates to ensure high confidence in the concentration-dependent effects. For the variant-specific experiments (Figure 3, n=3), after establishing HY's potent activity in the initial experiments, we proceeded with variant testing using n=3, which was sufficient to detect the substantial effect sizes observed (>85% inhibition at 20 nM) while maintaining statistical significance (p<0.05). Both sample sizes provided high reproducibility with low variability (as reflected in the standard deviations), providing confidence in the reliability of our findings across different experiments. Working with multiple SARS-CoV-2 variants in BSL-3 conditions imposes significant logistical and resource limitations. Each variant requires separate handling, and experiments with multiple variants must be carefully coordinated. Our statistical analysis confirmed that n=3 was sufficient for the variant experiments, as demonstrated by the statistically significant results and clear effect sizes observed. Increasing the sample size would likely reinforce the existing conclusions without substantially changing the overall findings. We have revised the figure captions to better explain our rationale for the experimental design and number of replicates used in each experiment, providing readers with clearer understanding of our methodological decisions.

Comment: Figure 3 should replaced in a new page 

Response: We appreciate the suggestion. Figure 3 (updated to figure 4 in the new version of the manuscript) position has changed.

Comment: Figure 4: Figure 4. Combination of HY with RDV and NMV. Antiviral activity of HY against SARS-CoV-2 in vitro, in combination with RDV (A), NMV (B), and RVD+NMV (C). SARS-CoV-2 infected Vero E6 cells were submitted to different treatment schemes, by using HY at 1, 10, and 100 nM and/or RDV (2000 nM) and NMV (4000 nM). The quality of this figure is very poor I can not read anything.

Response: We thank for this pertinent suggestion. We have included a figure with more quality. It is important to highlight that the jpeg format of the figures were also sent to the journal to improve quality and readability.

Reviewer 4 Report

Comments and Suggestions for Authors

The article titled “Hypericin suppresses SARS-CoV-2 replication and synergizes with antivirals via dual targeting of RdRp and 3CLpro” from Souza et al. presents an interesting study providing a combinatorial approach consisting of hypericin and other antivirals against SARS-CoV-2 infection.

However, several aspects require clarification and refinement to strengthen the overall impact, especially in discussing the results.

  • In lines 56-57, the statement that “high levels and replication in the upper respiratory tract favors the efficacy of antiviral therapy” is misleading and, as phrased, incorrect. A more accurate way to express this would be to clarify the concept that, when the SARS-CoV-2 viral load is high, the antiviral therapy tends to be more effective in reducing viral replication.

  • Line 78: the claim that the drug molnupiravir is an approved treatment for COVID-19 is inaccurate. According to FDA statement, molnupiravir is only authorized for emergency use when no other approved treatments are available or suitable. The authors should reconsider how they present this information (https://www.fda.gov/news-events/press-announcements/coronavirus-covid-19-update-fda-authorizes-additional-oral-antiviral-treatment-covid-19-certain).

  • Line 142: it would be more appropriate to use the term “compound” rather than “drug” when referring to HY, as it is not currently an approved pharmaceutical agent.

  • Paragraphs 4.5 to 4.7: These sections require a deep rephrasing to enhance clarity:

    First, the explanation of the time-of-addition assays is unclear, especially for readers unfamiliar with the methodology. I would suggest describing each assay separately highlighting their differences.

    2. This study tested HY at concentrations of 2000 nM and 200 nM. The reasons for the choice in each assay should be explained.

  • Each paragraph in the methods section should be separated for readability.

  • In the results section, it would be helpful to provide additional data to clarify the concentrations of NDV and RDV when used alone or in combination with HY.

  • In Fig.1A, the statistics is absent for certain concentrations. This should be corrected.

  • In lines 307-310, the description of the assays is confusing and should be rewritten for better clarity.

  • In the Fig.3 the statistics is absent in the pre-treatment–cell bar.

  • Line 413: correct “Hy” to “HY”.

  • Line 469: clarify the reference to “2 reports”.

  • Lines 483-485: a clearer statement should be made regarding the HY mechanism of action which, by binding to conserved residues rather than mutation-prone sites, is unlikely to be compromised by viral evolution.

  • The authors' interpretation of results in the discussion is somewhat unclear. The data suggest that HY is effective both in co-treatment assays and when pre-incubated with the virus, as well as when added post-infection. However, the authors seem strongly focused on the hypothesis that HY blocks viral fusion events in this work. This assumption forms the basis for the metadynamics simulations, yet it is not fully supported by other aspects of the study.

To strengthen the discussion: If metadynamics simulations are crucial to the findings, the importance of the fusion-blocking mechanism should be articulated and compared to other studies.

Author Response

Reviewer 4

Comment: The article titled “Hypericin suppresses SARS-CoV-2 replication and synergizes with antivirals via dual targeting of RdRp and 3CLpro” from Souza et al. presents an interesting study providing a combinatorial approach consisting of hypericin and other antivirals against SARS-CoV-2 infection.

However, several aspects require clarification and refinement to strengthen the overall impact, especially in discussing the results.

 Response:
Before addressing the reviewers' specific questions and comments, we would like to clarify an important point identified during a thorough revision of the manuscript’s methodology section, which was guided in part by the reviewers’ valuable feedback. We discovered a typographical error in the reported concentrations of the compound used in several experiments, specifically a shift of one decimal place (or one log unit). The text and associated figures have been corrected accordingly to reflect accurate values.

The following adjustments were made and marked in red throughout the text:

In the dose–response experiments, the correct concentration range used was 0.002–200 nM (corrected at methodology section and Figure 1A).

The corrected ICâ‚…â‚€ value for hypericin is 0.35 nM (corrected at results section and Figure 1B).

In the variant experiments, the concentrations used were 2 and 20 nM of hypericin (corrected at results section and Figure 2).

The time-of-addition experiment was correctly reported in the original version.

The dual-treatment experiment was also correctly reported and remains unchanged.

We appreciate the reviewers’ input, which led to this important clarification, and we believe the revised version more accurately represents the experimental design and results.

Comment: In lines 56-57, the statement that “high levels and replication in the upper respiratory tract favors the efficacy of antiviral therapy” is misleading and, as phrased, incorrect. A more accurate way to express this would be to clarify the concept that, when the SARS-CoV-2 viral load is high, the antiviral therapy tends to be more effective in reducing viral replication.

Response: We thank the reviewer for this important correction regarding the relationship between viral load and antiviral efficacy. We agree that our original phrasing was imprecise and could be misinterpreted. We have revised the statement to more accurately reflect the current understanding of COVID-19 treatment timing and efficacy, and changed the introduction section accordingly.

Comment: Line 78: the claim that the drug molnupiravir is an approved treatment for COVID-19 is inaccurate. According to FDA statement, molnupiravir is only authorized for emergency use when no other approved treatments are available or suitable. The authors should reconsider how they present this information (https://www.fda.gov/news-events/press-announcements/coronavirus-covid-19-update-fda-authorizes-additional-oral-antiviral-treatment-covid-19-certain).

Response: We thank the reviewer for this important correction regarding the regulatory status of molnupiravir. We acknowledge that our original statement imprecisely characterized the approval status of COVID-19 antivirals, particularly molnupiravir. We have revised the text to accurately reflect the current regulatory status of each antiviral agent and changed the introduction section accordingly. This revision provides a more accurate description of the regulatory status of each antiviral agent, clearly distinguishing between full FDA approval and Emergency Use Authorization, while highlighting the specific limitations on molnupiravir's authorization. The revision also maintains the important point about monoclonal antibodies while ensuring accuracy regarding the approval status of all mentioned therapies.

Comment: Line 142: it would be more appropriate to use the term “compound” rather than “drug” when referring to HY, as it is not currently an approved pharmaceutical agent.

Response: Thank you for the suggestion. We agree that “compound” is more appropriate in this context, as HY is not an approved pharmaceutical agent. We have revised the manuscript accordingly and replaced “drug” with “compound” throughout the text.

Comment: Paragraphs 4.5 to 4.7: These sections require a deep rephrasing to enhance clarity: First, the explanation of the time-of-addition assays is unclear, especially for readers unfamiliar with the methodology. I would suggest describing each assay separately highlighting their differences.

Response: Thank you for your valuable comment. We agree that the original section lacked clarity for readers unfamiliar with time-of-addition assays. In response, we have deeply rephrased the cited paragraphs, describing each experimental condition separately and clarifying the rationale and interpretation of each assay. These revisions aim to enhance comprehension and provide a more structured explanation of HY’s antiviral mechanism.

Comment: This study tested HY at concentrations of 2000 nM and 200 nM. The reasons for the choice in each assay should be explained.

Response: Thank you for raising this point. We have clarified in the Results section the rationale behind the selection of all tested concentrations of HY. Specifically, the concentrations of 2 nM and 20 nM were selected for experiments with viral variants based on prior dose–response data, corresponding approximately to the ICâ‚…â‚€ and near-IC₉₀ values, respectively, to assess antiviral efficacy across a physiologically relevant and non-cytotoxic range. For the time-of-addition assays, we used a concentration of 200 nM, which induces maximal antiviral effect without cytotoxicity, to better elucidate the mechanisms of action of HY.

Comment: Each paragraph in the methods section should be separated for readability.

Response: Thanks for the suggestion. We have performed this separation of the paragraphs for the whole text to increase readability.

Comment: In the results section, it would be helpful to provide additional data to clarify the concentrations of NDV and RDV when used alone or in combination with HY.

Response: We thank for this pertinent suggestion and we have included more detailed data on this specific question.

Comment: In Fig.1A, the statistics is absent for certain concentrations. This should be corrected.

We appreciate this important suggestion. Therefore, we have included statistical analysis results for all the data points related to the HY concentrations in the manuscript at Figure 1 of the Results section.

Comment: In lines 307-310, the description of the assays is confusing and should be rewritten for better clarity.

Response: Thank you for your observation. We agree that the original sentence lacked clarity. The description of the assay outcomes in respective lines has now been revised to improve readability and precision. We clarified the distinction between the treatment conditions and explicitly highlighted the differences in their effects on viral replication.

Comment: In the Fig.3 the statistics is absent in the pre-treatment–cell bar.

Response: We also appreciate this relevant suggestion. Therefore, we have included statistical analysis results for all the data points related to the HY concentrations in the manuscript at Figure 3 (updated to Figure 4) of the Results section.

Comment: Line 413: correct “Hy” to “HY”.

Response: Thank you for the suggestion. We have revised the manuscript accordingly and replaced “Hy” to “HY”.

Comment: Line 469: clarify the reference to “2 reports”.

Response: We thank for this pertinent suggestion. Therefore, we have included in the Discussion section that information regarding the “2 reports” refer to our group’s previous report published and the current one.

Comment: Lines 483-485: a clearer statement should be made regarding the HY mechanism of action which, by binding to conserved residues rather than mutation-prone sites, is unlikely to be compromised by viral evolution.

Response: Thank you for the suggestion. We have revised the sentence in the mentioned lines to more clearly state that HY’s mechanism of action targets conserved viral regions rather than mutation-prone sites, which may explain its maintained antiviral efficacy across SARS-CoV-2 variants. The revised version is now clearer in emphasizing this point.

Comment: The authors' interpretation of results in the discussion is somewhat unclear. The data suggest that HY is effective both in co-treatment assays and when pre-incubated with the virus, as well as when added post-infection. However, the authors seem strongly focused on the hypothesis that HY blocks viral fusion events in this work. This assumption forms the basis for the metadynamics simulations, yet it is not fully supported by other aspects of the study. To strengthen the discussion: If metadynamics simulations are crucial to the findings, the importance of the fusion-blocking mechanism should be articulated and compared to other studies.

Response: We thank for this pertinent comment. We have performed a complete change of the discussion section to better clarify and increase our arguments regarding mechanism of action and correlation with metadynamics. Although the co-treatment and virucidal assays strongly support the hypothesis that HY may inhibit viral entry by interfering with the fusion process, the significant reduction in viral replication observed during post-infection treatment indicates that HY can also act at intracellular stages of the viral life cycle. This dual activity profile suggests a pleiotropic antiviral mechanism, whereby HY may target multiple steps in the SARS-CoV-2 replication cycle. In this context, we employed metadynamics simulations specifically to explore one of the proposed mechanisms—HY interference with the viral proteins related to replication steps intracellularlu. Importantly, our simulation approach focused on the RdRp and 3CLpro proteins. While these findings are not intended to exclude other mechanisms of action, they provide molecular-level support for the one of the hypothesis.

Round 2

Reviewer 1 Report

Comments and Suggestions for Authors

The improvements made to the manuscript are remarkable, and I recommend acceptance.

Reviewer 2 Report

Comments and Suggestions for Authors

The authors have now addressed all the comments raised on the initial submission.

Reviewer 4 Report

Comments and Suggestions for Authors

The article has been deeply revised by authors and it is now suitable for publication.